# Development of an Innovative Online Dietary Assessment Tool for France: Adaptation of myfood24

**DOI:** 10.3390/nu14132681

**Published:** 2022-06-28

**Authors:** Anaïs Hasenböhler, Lena Denes, Noémie Blanstier, Henri Dehove, Nour Hamouche, Sarah Beer, Grace Williams, Béatrice Breil, Flore Depeint, Janet E. Cade, Anne-Kathrin Illner-Delepine

**Affiliations:** 1Dietary Assessment Ltd., Nexus, Discovery Way, The University of Leeds, Leeds LS2 3AA, UK; s.l.beer@myfood24.org (S.B.); g.williams@myfood24.org (G.W.); 2Institut Polytechnique UniLaSalle, 19 rue Pierre Waguet, 60026 Beauvais, France; lena.denes@etu.unilasalle.fr (L.D.); noemie.blanstier@etu.unilasalle.fr (N.B.); henri.dehove@etu.unilasalle.fr (H.D.); nour.hamouche@etu.unilasalle.fr (N.H.); 3Institut Polytechnique UniLaSalle, Université d’Artois, ULR 7519, Equipe PANASH, 19 rue Pierre Waguet, 60026 Beauvais, France; beatrice.breil@unilasalle.fr (B.B.); flore.depeint@unilasalle.fr (F.D.); 4Nutritional Epidemiology Group, School of Food Science and Nutrition, University of Leeds, Leeds LS2 9JT, UK; j.e.cade@leeds.ac.uk

**Keywords:** myfood24, nutritional epidemiology, web-based 24 h dietary recalls, database, food portion pictures, dietary assessment

## Abstract

myfood24 is an innovative dietary assessment tool originally developed in English for use in the United Kingdom. This online 24 h recall, a tool commonly used in nutritional epidemiology, has been developed into different international versions. This paper aims to describe the creation of its French version. We used a consistent approach to development, aligned with other international versions, using similar methodologies. A nutritional database (food item codes, portion groups and accompaniments, etc.) was developed based on commonly used French food composition tables (CIQUAL 2017). Portion sizes were adapted to French dietary habits (estimation, photographs of French portion sizes, assessment of the photograph series and their angle (aerial vs. 45 degrees)). We evaluated the new tool, which contained nearly 3000 food items with 34 individuals using the System Usability Scale. We validated the French food portion picture series using EFSA criteria for bias and agreement. The results of the picture evaluation showed that the angle with which photos are taken had limited impact on the ability to judge portion size. Estimating food intake is a challenging task. Evaluation showed “good” usability of the system in its French version. myfood24 France will be a useful addition to nutritional epidemiology research in France.

## 1. Introduction

Diet is an important lifestyle-related risk factor for chronic diseases, but it raises many challenges to be properly measured. All self-reported or objective dietary assessment tools and methods are subject to measurement error [1]. Choosing the appropriate method is, therefore, a complex decision based on the objective of the data collection and the target group (individual or population), considering the competing demands of accuracy and practicality.

In nutritional epidemiology, food frequency questionnaires lack accuracy because of recurrent problems such as remembering what one ate, misconception of portions, or limitations of food lists and food composition tables, as reviewed by Eldridge et al. [1]. Thus, 24-hour dietary recalls (24HDRs) may be more accurate [2], but respondents need to complete more than a day’s recall to allow within-person variation to be estimated. The European Food Safety Authority (EFSA) considers 24HDRs to be a gold standard in short-term human food consumption surveys in Europe to estimate habitual dietary intake on an individual level [2].

Technologies have undergone tremendous development over the past decades. Their evolution has led to the emergence of new tools in dietary assessment: methodology evolved from an all-paper method in the 1980s to web-based ones [3]. When using paper-based methods, large-scale studies may collect partial and less detailed information due to missing or incomplete data. Technology enables dietary assessments to automate prompts and requires less financial and human resources in this setting.

myfood24 uses technology to help data collection using a 24HDR [4]. Online solutions such as myfood24 may enable detailed data to be obtained more frequently at a lower cost and require less manual work for both professionals and participants. Little is known about the dietary data quality and the challenges the participants face when quantifying their intake with these new methods.

myfood24 is a quick and easy-to-use online dietary assessment tool for research, teaching, and healthcare settings. It has potential for use in large-scale studies. It is already available in the United Kingdom (UK), Australia, the Caribbean, Germany, Denmark, Norway, the Middle East, Peru, Uganda, and the United States of America. It is mobile-optimised, which eases data collection for both participants and professionals. This article explains how we developed the French version of myfood24. During this process, focus has been put on the quantification of food intake using food portion pictures.

## 2. Materials and Methods

The development of the French version of myfood24 can be divided into three main steps:Developing the food composition database;Adapting portion sizes to the French dietary habits;Evaluating the French version.

This methodology aligns with the tool’s other versions. In particular, the development of the food database was needed first considering the already existing structure of myfood24, which is an open-ended and online dietary assessment method. More details on the different steps can be found below and are illustrated in the Appendix A.

### 2.1. Development of the French Food Composition Database

The first step of the database’s development was to identify usable French food composition data. These were the French national food composition database, CIQUAL 2017 [5], and part of J.-P. Blanc’s book of food back-of-pack nutrition information [6]. No data from myfood24′s other language versions have been used, as only the generic items would have been relevant, and these were already in the CIQUAL tables.

The second step was to remove all items that were not likely to be consumed due to their nature or processing. For example, to avoid confusing the user, “raw beetroot” has been deleted as it is not commonly consumed in France.

The third step was to remove all non-specific combined foods present in the CIQUAL tables (*n* = 50) when their components were quantifiable or more precise options were available. Composite dishes that included the word “average”, such as an “average sandwich” have been removed to encourage users to select individual ingredients or more precise options. For example, the participant should select a more precise sandwich (e.g., “sandwich baguette, jambon, beurre”), or quantify all of the ingredients in their sandwich. This will be different between individuals, potentially improving accuracy, instead of choosing an average sandwich (“sandwich (average)” [5]) where the components would be unknown to the participant.

The fourth step was to change the technical descriptions to more comprehensible ones. For example, the CIQUAL food name “fermented milk or dairy product like yoghurt, flavoured, sweetened, with bifidus” has been changed into “sweetened flavoured yoghurt with bifidus” which is more understandable for the general public.

The fifth step was to add synonyms and common misspellings to help/assist the search function. For example, for a pain au chocolat a synonym “chocolatine” was added.

The sixth step was to add common accompaniments (e.g., sugar with coffee). If the participant adds an item linked to a group of common accompaniments, the elements of this group will generate an on-screen prompt asking them if they may have forgotten something.

The seventh and final step was to add information on the type (branded/generic) and food category of each item. These help the participant filter the search results.

Codes were assigned to each food/drink item, giving information about their food group, and food subgroup. As classification methods can be different between countries and continents, we used the Food and Agriculture Organization of the United Nations/World Health Organization Global Individual Food Consumption Data Tool (FAO/WHO GIFT) classification [7].

Selected items from J.-P. Blanc’s book of food items’ back-of-pack nutritional data [6] were copied manually from other dietary recall software an item at a time. Due to the time-consuming nature of this work, these items were selected before being input; as such, they do not represent the entire branded database. Food selected for addition from J.-P. Blanc’s book were chosen to increase the diversity within some of the CIQUAL subgroups and make the database more representative of the French food and drink market.

### 2.2. Adaptation of Food Portion Sizes to French Dietary Habits

#### 2.2.1. Food Portion Estimation

Each food item has been assigned a portion description to help the participant estimate their intake. Portion suggestions will be prompted on-screen, with pictures, if available, and the participant will choose the portion size picture closest to what they ate or write the exact weight (in grams) of the item.

Data were also available on usual portion size, as consumed, from the Food Observatory (Observatoire de l’Alimentation (OQALI)) website [8]. OQALI monitors the food offered on the French market and collects portion size information on labels when indicated by the manufacturers. These are available by food group. OQALI did not provide information for all groups, so when information was not available, portion sizes were collected on French supermarkets’ websites and Open Food Facts France [9]. These additional portion sizes were also added to the database.

#### 2.2.2. Food Portion Photographs

The list of new pictures was limited to 20 foods that are commonly consumed in France. Items were selected according to their food type and shape. Forty further food photograph portion series were also included; these used the same pictures as myfood24 UK offers its participants.

These 20 new French items’ standard portion weights, which were found on the internet, were compared to the food photographs of the 2002 French Food Atlas, SU.VI.MAX [10], for reference.

Four to seven pictures of each food item were taken to add to myfood24. The French pictures protocol was developed by merging the one used by myfood24 and the one from the International Agency for Research on Cancer (IARC) [11].

The final protocol was as follows:Installing the photo box: switching the lights on, setting a white background, the tripod, and cutlery (see Figure 1);

2.Installing a scale, taring it with an empty plate;3.Weighing and recording of the food items at ambient temperature (to avoid condensation and evaporation);4.Photo taking: taking photos from the smallest to the biggest portion, placing food from the left side to the right of the plate as the portion becomes bigger. For some food items, pictures had to be taken from the biggest to the smallest (e.g., baguette). For each portion, two photos were taken: one aerial (overhead) and one angled (45°);5.Editing of the pictures.

Once all 168 pictures (eighteen series of four portions, one series of five portions, and one series of seven portions) were taken and edited, a selected sample was evaluated as described below. All photos were then entered into the prototype of the French version of myfood24.

#### 2.2.3. Food Portion Evaluation

The two objectives of this evaluation were to assess the food photograph series and estimate the impact of the angle of the pictures (aerial vs. 45°).

The protocol for this evaluation was based on the previous work of Subar et al. [12] for food item selection and design, and that of the EFSA [13] for the statistical approach.

An email invitation was sent to the UniLaSalle university employee mailing list asking for volunteers and providing them with different time slots for which participants could register. In-person advertisement also proved useful. The assessment was divided into two steps: onsite evaluation and virtual questionnaire. At the time of the evaluation, due to COVID-19, people only worked on-site three days a week. Therefore, for convenience, only the first part of the assessment required their presence on site. When this evaluation took place, only seven people could be in the same room without a mask due to safety protocols, each being separated by at least one meter from the others. Thus, the participants were divided into four groups.

On the first day, the participants came on-site and helped themselves at a self-serve buffet. Five food categories were available: amorphous foods, single-unit foods, small pieces, spreads, and shaped foods [12] (see Table 1). As only one food portion series was required for spreads and shaped foods, the first two groups did not evaluate these categories.

The participants’ plates were weighed each time they helped themselves to one food and again at the end of the meal if there was anything left.

On the second day of the evaluation, the participants received an email with their identification number and the link to an online form (Google Forms) and were told to answer it within the day. Each group had a specific form. The participants were asked to answer the questionnaire from their phone for improved layout.

The forms were divided into four parts:Presentation and complete explanation of the evaluation (participants did not know they had to evaluate their intake on the first day—they only discovered this on the second day);Identification number;Test: photos were displayed for each food item from the menu offered on the first day. The participants had to choose the closest estimation to what they ate: the exact amount of the picture or an intermediate amount (e.g., tick the box “between picture 1 and 2”) or tick the box “not consumed”;Personal information: the participants had to enter their information (sex, age, socio-professional category, smoking status, height, weight, colour blindness, and nutritional background).

The questionnaire’s results were analysed by comparison against EFSA criteria [13] and compared to the results of other studies [12] to evaluate the portion pictures.

Statistics were run on the collected results: real weights and selected weights.

First, the absolute difference was calculated between these two different weights for each respondent (in grams, percentages, and increment of portions) according to the following Equation (1):difference = |real weight − declared weight|.(1)

Then, an analysis of variance was run to test whether the differences in weights by image type (aerial or angled) were significant or not.

The respondents’ degree of agreement (with each other) was measured by calculating the Intraclass Correlation Coefficient (ICC). The criterion has been calculated for real and declared consumptions per food by image type. Finally, the percentages of people who were able to evaluate their consumption per food within a 10% margin were calculated. The 20 new sets of aerial pictures were added to myfood24 France.

### 2.3. Evaluation of the Prototype

The evaluation was designed to assess the technical issues, usability and user-friendliness, participant interface, search for food and other features, completion time, and acceptance, and allowed the participants leave personal comments and suggestions [14].

A new group of participants was recruited in March/April 2021. The participants had to match all of the seven following criteria to join the study [14]:Being over 18 years old;Speaking French fluently;Having a regular high-speed Internet access;Having a valid email address;Being stable in bodyweight (no diet) during the study;Being free from any metabolic disease;Being willing to maintain their current dietary and activity behaviours during the whole study.

The participants completed three 24HDRs in four months using myfood24 France, answered a questionnaire, and participated in a focus group.

The participants completed their first recall a week after a presentation of myfood24 and a quick explanation of the main features. The presentation showed the search function, search filters, portion size selection, and recipe builder.

The participants were randomly assigned to complete myfood24 France to represent all days of the week (Monday to Sunday): two weekdays and one day during the weekend (Saturday or Sunday). The recall requests were sent by email.

Participants who completed the three dietary surveys were emailed a questionnaire on the prototype’s acceptability and usability. They were also invited to participate in a focus group to discuss these points.

## 3. Results

### 3.1. Development of the Nutritional Database

Figure 2 summarises the results of the development of the French nutritional database.

The 2017 official French food composition tables were made up of 2807 food and drink items. Nine of CIQUAL’s subgroups (salad-based dishes, other meat products, dairy products and desserts, jams, ice creams, sorbets, frozen desserts, cooking aids, and food for particular nutritional uses) that were deemed not to adequately reflect the French market were enhanced with 225 selected items for J.-P. Blanc’s book’s database, for a total of 3032 items. After further steps detailed in Figure 2, the final database included 2981 food and drink items, 28 accompaniment groups, and 211 portion groups.

### 3.2. Acceptance of Food Portion Sizes

#### 3.2.1. Respondents’ Characteristics

Thirty-four respondents out of 428 subjects invited were recruited to evaluate the photos (cooperation rate = 9%). In total, 82% of them were women (*n* = 28) (see Table 2). Most of the respondents (38%) were between 25 and 34 years old. Concerning the social-professional category of the respondents, 65% of them were managers or had higher education professions.

#### 3.2.2. Mean Differences between Weighed and Reported Portion Sizes by Image Type

Table 3 summarises the mean weighed portion by image type in grams, mean absolute weight differences between weighed and reported amounts in grams and percentage. Each aerial series was evaluated by eight people and each angled one by nine.

The aerial pictures were slightly more accurate than the angled photos for most foods (see Table 3). Only lentils showed a significant difference between aerial and angled photos (*p* = 0.02).

#### 3.2.3. Bias of Food Portion Size Pictures and Agreement among Respondents

The difference between weighed and estimated measures the bias of a food portion picture series. Table 4 summarises the results concerning the EFSA difference criterion (expressed as multiples of portion increments) [13]. For example, the bias of aerial Greek salad pictures was 2.26 × 50 = 113 g: the participants underreported their intake by 113 g for each portion picture. No matter their portion, the participants underreported their consumption by 113 g (out of their weighed portion) on average.

The Greek salad and yoghurt series show a large bias (difference greater than 1.00). The angled lentil and cherry pictures as well as the angled and aerial photos of baguette, peach, and tapenade show a moderate bias (difference between 0.5 and 1.00). The aerial lentil and cherry series and both angled and aerial pictures of cheese show a low bias (differences between 0.25 and 0.5). Both series of sausage pictures show no bias (difference less than 0.25).

The ICC measures the degree of the agreement the respondents have about a picture series [13]: the higher the ICC, the more the participants perceived pictures in the same way. Table 4 summarises the results.

The Greek salad shows a moderate agreement (ICC between 0.41 and 0.60) as well as the aerial peach pictures. The angled cherry, angled and aerial yoghurt, aerial peach, angled tapenade, and angled cheese series show a substantial agreement (ICC between 0.61 and 0.80). The lentils, baguette, sausages, aerial cherry, aerial tapenade, and aerial cheese series show an almost perfect agreement (ICC between 0.81 and 1.00).

#### 3.2.4. Percentage of Foods Estimated within ±10% of Measured Weight

Estimating one’s food consumption can be a challenging task. This is what the four last columns of Table 4 show. The participants’ best estimates were units (cherries, sausages). The respondents faced more difficulties for the foods to which they helped themselves several times (cheese, baguette) as well as spreadable food (tapenade). The percentage of people who could accurately estimate their consumption was calculated (±10%) [12].

### 3.3. Tool Evaluation Results

Forty-six participants were recruited in April/May 2021 for the usability evaluation of the tool. Thirty-one completed all three 24HDRs with myfood24′s French version, 26 of them completed the questionnaire, and 16 participated in the focus groups.

#### 3.3.1. Survey Results

The survey results were assessed through the questionnaire that the participants had to answer after their third recall with the tool.

Table 5 shows the System Usability Scale (SUS) scores for myfood24 France. The median SUS score was good (73, Q1–Q3: 66–84), which indicates that myfood24 France is usable; however, the mean value was lower (68) and corresponds to an acceptable system [15,16]. The median SUS score of adults (63, Q1–Q3: 46–76) over 30 years old (y.o.) was 10 points lower than that for young adults (73, Q1–Q3: 63–78). The median SUS score for women (73% of the participants) was 73 and 60 for men.

Twenty-four users did not encounter any difficulty loading and opening the application and two encountered difficulties between one and four times. Other technical issues were noted by three users when filling out the app: hitting the browser’s back button is impossible (*n* = 1) or deletes data (*n* = 1), and some foods are impossible to find (*n* = 1) as they were not available and/or under a different name than the used search terms.

The majority of the participants agreed that the interface is uncluttered (n = 24) and intuitive (*n* = 23). The messages guiding the user through the completion process were judged useful and just enough by 81% of the users. Moreover, 11% found them too numerous and 8% not numerous enough.

The search function of myfood24 was considered easy to use by 81% of the participants, but only 27% of them mentioned finding their food quickly. Participants who reported having difficulty finding the right foods stated two major causes, which were that “Some (or all) of (the) foods searched for were not available (brands)” (*n* = 17) and “Foods were not sorted in a relevant way” (*n* = 8). Most participants also stated that the database was lacking some items, that there should be more branded foods. They also reported that the high number of search results and their display in alphabetical order made finding the right item or ingredient a time-consuming task. The limited number of synonyms was also mentioned. Portion images were found to be useful by 92% of participants. The different functions of myfood24, such as the recipe builder, the ingredient list, and the recently added foods, were used by 46%, 23%, and 19% of the participants, respectively. In addition, 58% of the participants who used the recipe builder found it easy to use. The others found it tedious and time consuming to use and sometimes difficult to navigate.

#### 3.3.2. Focus Group Results

Sixteen of the participants in the evaluation study took part in a focus group. The majority of the participants were female (69%). The median age was 21 years (19–58). The participants were divided into five adults (>30 years old) and 11 young adults (<30 years old). Forty-four percent of participants reported having a background in nutrition or food science and 31% reported having previous experience with food consumption-recording applications and tools. The results collected through the focus groups were broadly consistent with the results of the questionnaire and thus support the general acceptability and points for improvement mentioned earlier.

The participants stated that they did not encounter difficulties in using the tool. Whether the mobile-optimised in-browser window or the computer version, the tool does not pose a problem in terms of being loaded and used. Three users highlighted the speed of the tool as a positive aspect, especially for reporting, which is obtained instantly after submitting the 24HDR.

All participants noted that the photos did not always correspond to the food they were looking for and that it would be more relevant to have specific photos to correctly estimate the portions. The messages included for accompaniments and commonly forgotten foods were deemed relevant, but sometimes too long, especially in the case of food accompaniments. The concept of prompts was appreciated, but the proposed accompaniments were too numerous and therefore discouraged reading.

The database was described by the participants as limited in relation to branded items and exotic foods, since they often had to choose a substitute. Many users found the recipe builder time consuming and preferred to search for a substitute food rather than create their own recipes. However, saved recipes can be reused for future recalls, which is an advantage for people who eat certain dishes on a recurring basis. One user stated that the recipe builder was a time saver for them afterwards. The portions were judged to be well-considered with lots of intermediate quantities and when images are available, a range of portions is pictured, which makes it easier to visualise and complete the diary.

According to the feedback given during the focus group, the application was considered user-friendly and practical in terms of navigation, but it was also described as limited in relation to the number of food items and for some people it was considered time consuming. The users were satisfied with the application, but three quarters of them would only recommend it after modifications.

#### 3.3.3. Time of Use

The average completion time was assessed through the questionnaire. One third of the users estimated that they spent between five and nine minutes filling out their 24HDR. The remaining users spent more time, with some (*n* = 3, 12%) estimating spending more than 30 min. The focus groups provided an opportunity to discuss the ideal amount of time to complete a food diary and the amount of time users could make available. Eighty-one percent of users would be willing to spend between 10 and 14 minutes filling it in. The time taken to complete a day’s intake by participants who reported having a background in nutrition or food science as well as those who reported having previous experience with a food consumption-recording app was not significantly different from the other users. Fifty percent of the respondents were willing to use myfood24 on a regular basis, and 57% of them once a week.

## 4. Discussion

### 4.1. Size of Database and Missing Data

This database is based on the CIQUAL 2017, which may have limited the food choice in myfood24 France. The enhancement of some CIQUAL subgroups is not sufficient to represent the French food market’s diversity. French cuisine is full of culture, heritage, traditions, and pride, and is constantly evolving. It is so complex and rich that the United Nations Educational, Scientific and Cultural Organization (UNESCO) “inscribed [the French gastronomic meal] in 2010 […] on the Representative List of the Intangible Cultural Heritage of Humanity” [17]. As manufacturers launch new products regularly, a food database will always be missing data. The French version does not cover as many items as the UK and German versions, which contain information on numerous branded items (UK: over 88,000 branded items). Over 10,000 branded food items and 60 traditional recipes have now been added to myfood24 France since this evaluation and in response to feedback. Other countries’ food items were not included in the French database as their information depends on many factors (e.g., geography, local practices).

### 4.2. Food Portions

#### 4.2.1. Mean Differences between Weighed and Reported Portion Sizes by Image Type

Due to the lack of publications on this topic, the difference of perception on technology-based methods according to a picture’s angle could not be compared to other work. Further studies would be needed to demonstrate that the perception of food portion sizes changes according to the angle at which a picture was taken, but our preliminary data suggest that the impact is limited in most food groups tested.

#### 4.2.2. Bias of Food Portion Pictures

The Greek salad pictures did not reflect amounts of food large enough to reflect the participants’ actual consumption; however, the portions used were the same as those in the French Food Atlas [10]. Adding larger portions of food could correct this bias. The evaluation form’s labels of the yoghurt series (“1”, “2”, “3”, “4”) may have confused the respondents’ understanding (one whole yoghurt was not labelled “1”, but “3”). The peach series was affected by the pictures’ labels in the same way as yoghurt’s. The baguette, peach, tapenade, and cherries and lentils angled picture series show a moderate bias. French people eat a considerable amount of bread, and help themselves to further portions, making it difficult for them to estimate global amounts. Evaluating one’s consumption of spreads (e.g., tapenade) is challenging, even more so since the difference between portions is small (5 g). Impacted by the photo angle, the lentil angled pictures show a moderate bias. The cheese, aerial lentils, and cherries pictures show a low bias. The respondents were observed to only help themselves to cheese once instead of several times (e.g., bread). Thus, the pictures seem appropriate for the consumption estimation for these foods. The sausage food picture series shows no bias. It was easy for the respondents to recall the number of sausages they ate, and the pictures’ labels matched the number of represented units. This concords with others’ findings [12] stating that foods in individual units are easier to assess.

#### 4.2.3. Agreement among Respondents about Picture Series

The agreement with EFSA criterion shows that all respondents viewed and perceived the portions represented in the photographs in the same way in most cases. Overall, the agreement (ICC) was better for the aerial picture series.

#### 4.2.4. Percentage of Foods Estimated within ±10% of Measured Weight

The percentages of foods estimated within 10% of the measured weight were higher in 2020 in France than previously reported in the United States in 2010 (respectively 33% vs. 15% for aerial pictures, and 21% vs. 18% for angled ones) [12]. Several factors could explain this difference.

This French study is much more recent. People may now be more familiar with the technology supporting food consumption evaluation. The two countries also have different food cultures. The French view food as an experience and associate it with enjoyment and tend to eat smaller portions; they “are governed more in their eating and food decisions by optimising pleasure, following tradition, being moderate, valuing quality over quantity, and joy over comforts” [18]. As portion sizes tend to be bigger in the USA than in France, French people may pay more attention to the portions they eat and thus may tend to evaluate their intake more accurately. The French enjoy food as an “experience rather than […] a health behaviour” [19]. Because of COVID-19, the French study’s participants were recruited on a university campus, whereas the American study represented all ages and education levels. The similarity of participants’ backgrounds regarding age and education (recruitment on university campus) in the French study may have affected the results of food intake estimation. The two studies also called for different devices. In France, the participants answered the questionnaire from their smartphone, whereas in the USA, they had to answer on a computer (computer software program). This may have affected the participants’ perception of the pictures as the screen sizes are different, and a smartphone is often held horizontally (180°) while a computer screen is vertical (90°).

### 4.3. Alternatives to Portion Pictures

A few alternatives to portion pictures exist. One is weighing everything that is consumed. However, such a time-consuming task may change the participant’s eating behaviour [20] and interfere with the reporting quality. For instance, the complex approach to measurement is likely to impact on reported data detail as subjects are less motivated and may drop out. Household measures or descriptions could also be used (e.g., a slice of bread). Estimates using household measures “can be improved with training but the effects may deteriorate with time” [20]. Studies have shown that portion pictures are more useful than household measures to evaluate the amounts consumed [20]. The participant could also take a picture and then image recognition would be performed either by a trained person or through artificial intelligence (AI). The former would review the participant’s plate picture on a computer screen by comparing them to reference portions and will then enter estimates manually [3]. If this solution eases the respondent’s burden, it requires someone else to do the estimation, which is costly. AI would recognise the food and its quantity from a picture. The accuracy with which AI can identify certain foods and portion sizes is high [21,22,23,24,25]; however, this method is still not perfect and improvements are still possible. Additionally, picture upload means that the food recall is always prospective, hence, not a real recall.

### 4.4. Evaluation of Usability

myfood24′s UK, German, and French versions have a high and similar usability score (median SUS scores being respectively 80 (*n* = 20) [4], 78 (*n* = 92) [14], and 73 (*n* = 26)). The French study was smaller and its response rate lower, with only 26 (response rate: 62%) volunteers compared to the German study, which obtained 92 responses from 97 participants (response rate: 95%) [14]. As the participants were recruited on the UniLaSalle campus in Beauvais, not all socio-professional and age categories were represented. Most were students, between 18 and 25 y.o. This population may be more demanding and critical about apps and technology than others: they are much more familiar with it and use it on a daily basis. No elderly people, who may have more difficulties with technology literacy, tested the French version. Another bias is that 38% (*n* = 10) of our participants “received training in nutrition”. Four participants dropped out because of the time completion being judged too long. Yet, one French participant in three only spent five to nine minutes completing their diary. Comparing the results of other studies, the French actually had the quickest completion time (France: median 12 min (Q1–Q3: 7–22); Germany: median 15 min. (Q1–Q3: 9–23) [14]; UK: mean 19 min. (SD:7) [4]). The completion-time-induced dropouts concord with the findings of the UK study [4]: younger participants are more demanding and request a shorter completion time (11–18 y.o.: 10–15 min; above 65 y.o.: 20–30 min, 60 min if infrequent diaries).

### 4.5. Why Use an Online System Rather Than a Paper-Based One?

Technology and, more specifically, online systems can ease dietary assessments, including for the participants. In 2020, France counted 92% internet users in the country (83% inhabitants connecting daily). On average, that year, the French spent 19 hours on the internet per week [26]. Using an online system to assess diet now allows individuals to use systems they are familiar with and data are processed in real time [3]. Moreover, in 2020, no less than 84% of people had a smartphone (against 61% owning a computer) [26]. In 2019, 51% of French people preferred to use their smartphone as their first option to go on the internet, against 31% for computers [27]. The myfood24 online system is therefore easy to access, especially through its mobile-optimised version.

Smartphone and computer ownership is more common among younger adults. Education level and the location of individuals also affects access—87% of Parisian inhabitants owned a smartphone in 2020 compared to 77% in rural communities [26]. Technology literacy varies according to the gender, age, household size, education level, and location. For instance, people over 70 years old and people without diplomas would rather not use the internet in regard to public administration (e.g., reporting and payment of taxes) [27]. This resistance could also be associated with use of the internet in relation to dietary assessment.

### 4.6. Strengths and Limitations (Tool and Approach)

Pictures helped the respondents to estimate their intake. Only 20 picture series were taken due to limited access to campus as a result of the COVID-19 lockdowns. Only small groups of participants were allowed to take part in the evaluation. However, overall, 1889 of the 2981 items (63%) in the database were linked to portion pictures (20 French portion pictures and 40 original myfood24 series). The aerial portion pictures were marginally more helpful to the French population for their estimation of consumption (on a smartphone). The current database is not big enough to represent the diversity of the French food market. However, the database gives a good overview of food intake in France and features more items than the official food composition tables, and more branded items are currently being added to the database.

A standard approach was used to develop myfood24 France, similar to the development of other versions [14,28]. The items of the French version were also coded according to the FAO/WHO GIFT classification, which facilitates international studies. As the GIFT classification is based on EFSA’s FoodEx2 (European classification), many French-speaking researchers can use it without further training since they already use it in their work.

### 4.7. Recommendations

This is an exciting step forward for French dietary assessment, providing the first online food diary system. However, the database’s size should be increased. The database used in this work does not represent French cuisine’s diversity. Many commonly consumed items are missing. The French eat or drink a majority of branded items, yet they currently represent only 7.5% of the database tested in this paper. Adding such elements would be a considerable asset. Since this project, over 10,000 branded items and 60 recipes have since been added to the first version of myfood24 France. Adding more food picture series would be a good addition to the database. The 20 French portion picture series that have been taken help reflect the French dietary habits. So far, 28% (60 out of 211) of the portion groups used in myfood24 France are available with photos. Increasing the number of food picture series would help respondents even more in their food consumption estimation.

## 5. Conclusions

We have developed the first version and database for myfood24 France, a 24HDR method, which is compatible with other versions for international studies. In the version tested here, there were more than 3000 French foods in the database, and portion sizes were adapted to French dietary habits. Different approaches to the presentation of food portion pictures were tested. Overall, aerial photographs allowed a slightly better perception of portion sizes; however, the differences with angled photos were not big enough to be statistically significant (except for one item). The pictures were validated on small samples against EFSA criteria. The tool’s usability was evaluated and resulted in a “good” system usability. The current version still needs further validation (including with more diverse groups) and improvement, for example, with the search function, and more pictures would be needed. Since this first version, more than 10,000 items and 60 traditional regional recipes have been added to myfood24 France’s databases as well as 12 new portion picture series. myfood24 France can be used for nutritional assessment at the individual and population level.

## Figures and Tables

**Figure 1 nutrients-14-02681-f001:**
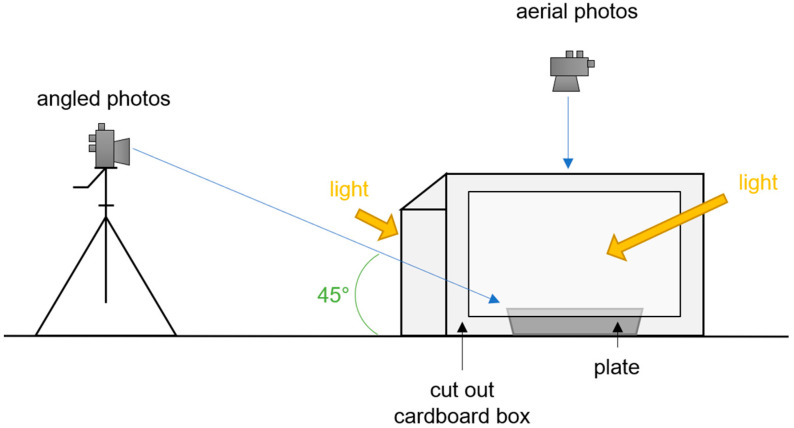
Schema of the installation of the photo box.

**Figure 2 nutrients-14-02681-f002:**
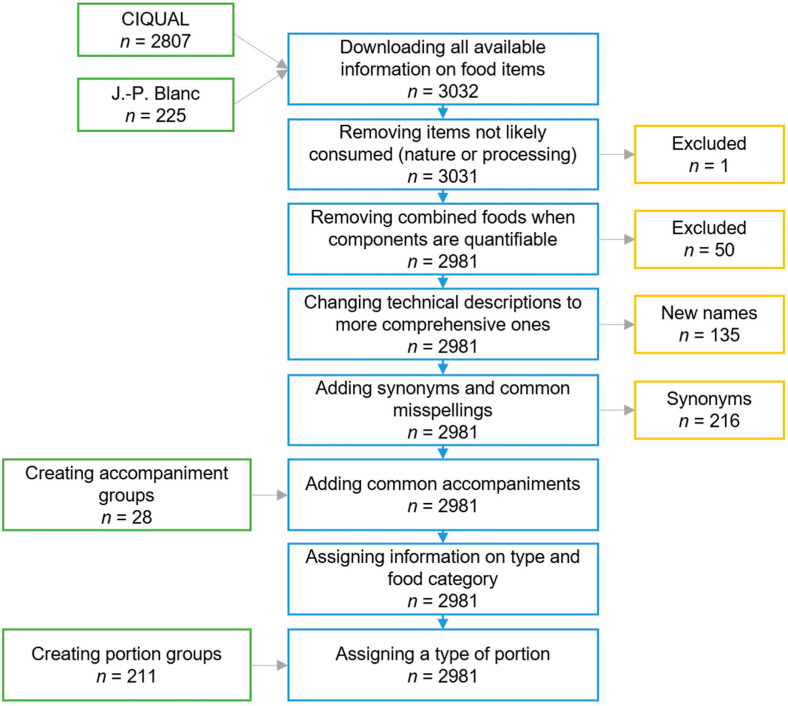
Database’s overall development methodology.

**Table 1 nutrients-14-02681-t001:** Menus of the photograph evaluation (June/July 2020).

	Group 1	Group 2	Group 3	Group 4
Number of participants	8	9	8	9
Type of evaluated pictures	Aerial	Angled	Aerial	Angled
Amorphous foods	Lentils		Greek salad	
Single unit foods	Sausages		Baguette	
Small pieces	Cherries and Yoghurts		Peaches	
Spreads			Tapenade	
Shaped foods			Camembert	

**Table 2 nutrients-14-02681-t002:** Participant characteristics (*n* = 34).

Characteristics		*n* (%)
Gender	Female	28 (82)
Age (years)	0–24	1 (3)
	25–44	20 (59)
	45–64	12 (36)
	65 and more	1 (3)
Socio-professional category	Craftsmen, traders, and entrepreneurs	1 (3)
	Managers and higher education professions	22 (65)
	Employees	5 (15)
	Students	2 (6)
	Intermediate professions	3 (9)
	Skilled workers	1 (3)
Body mass index (BMI)	Healthy weight	18 (53)
	Overweight	7 (21)
	Moderately obese	4 (12)
	Severely obese	1 (3)
	N/A	4 (12)

Thirty of the 34 participants reported their height and weight. Fifty-three percent of the participants were of healthy weight (*n* = 18), 21% were overweight (*n* = 7), 12% were moderately obese (*n* = 4), and 3% severely obese (*n* = 1) (see Table 2). None of the respondents was colour-blind. Six of them had nutritional training (18%).

**Table 3 nutrients-14-02681-t003:** Results of the photograph evaluation: comparison of mean differences of aerial and angled pictures.

Figure	Mean Differences								F-Ratios (*p* ≤ 0.05)	
	Aerial Photographs				Angled Photographs				F-Ratios	*p*-Value
	Mean Weighed Portion Size (g)	Mean Estimate Portion Size (g)	Mean Difference between Estimate and Weighed (g)	Mean Difference between Estimate and Weighed (%)	Mean Weighed Portion Size (g)	Mean Estimate Portion Size (g)	Mean Difference between Estimate and Weighed (g)	Mean Difference between Estimate and Weighed (%)		
**Amorphous foods**										
Greek salad	317	204	113	36%	227	179	57	25%	1.92	0.19
Lentils	109	117	15	14%	152	171	33	22%	6.29	0.02
**Single-unit foods**										
Baguette	61	82	18	29%	57	75	20	35%	0.05	0.83
Sausages	1.3	1.2	0	6%	2	1.8	0	6%	0.01	0.93
**Small pieces**										
Cherries	106	119	17	16%	136	134	32	23%	3.73	0.07
Yoghurt	94	64	37	40%	111	70	42	37%	0.08	0.78
Peach	116	79	40	34%	86	67	40	47%	5.85 × 10^−6^	1.00
**Spreads**										
Tapenade	8	7	3	36%	12	10	5	37%	0.58	0.46
**Shaped foods**										
Cheese	33	34	8	24%	23	31	13	55%	0.76	0.40

**Table 4 nutrients-14-02681-t004:** Results of photograph evaluation: comparison of image type with respect to bias, agreement, and estimation within 10% of measured weight.

Food Categories and Foods	Bias (EFSA Criterion [13])			Agreement (EFSA Criterion [13])		Percentage of Estimates within 10% of Measured Weight by Image Type	
	Increment of Portion Size	Aerial Difference (in Portion)	Angled Difference (in Portion)	Aerial ICC	Angled ICC	Aerial Photographs	Angled Photographs
**Amorphous foods**							
Greek salad	50 g	2.26	1.15	0.54	0.53	25%	33%
Lentils	50 g	0.29	0.66	0.85	0.87	50%	22%
**Single-unit foods**							
Baguette	30 g	0.58	0.67	0.95	0.89	0%	0%
Sausages	1 unit	0.06	0.06	0.97	0.98	75%	89%
**Small pieces**							
Cherries	50 g	0.33	0.63	0.92	0.75	50%	11%
Yoghurt	25 g	1.25	1.39	0.79	0.69	13%	22%
Peach	0.5 unit	0.80	0.80	0.64	0.60	50%	0%
**Spreads**							
Tapenade	5 g	0.60	0.90	0.89	0.75	38%	11%
**Shaped foods**							
Cheese	30 g	0.26	0.42	0.94	0.66	25%	22%

**Table 5 nutrients-14-02681-t005:** System Usability Scale (SUS) scores for a 24HDR with myfood24 France for the total sample and stratified by age group and gender.

	Total Participants	Employees (>30 y.o.)	Students (<30 y.o.)	Women	Men
Participants, *n*	26	7	19	19	7
SUS score median	73	63	73	73	60
Quartiles Q1–Q3	66–84	46–76	63–78	64–80	54–73
Extreme min-max	35–90	35–83	53–90	35–90	50–78

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
