# Peer review of "Development of an Innovative Online Dietary Assessment Tool for France: Adaptation of myfood24"

_nutrients, 2022, doi:10.3390/nu14132681_

Round 1

Reviewer 1 Report

The authors describe the process of adaptation to French circumstances of online 24-hour dietary recall questionnaire “myfood24”. The tool was originally developed in United Kingdom, however it is available and has been validated in some other countries as well.

 Developing the nutritional database, adapting portion sizes to the French dietary habits and evaluating the French version were the main steps undertaken for implementation of the tool in France. Each step has been described by the authors in details.

 In my opinion, the most interesting for the international audience is the part in which there is a comparison between real weight and declared weight, additionally presented by the types of foods and image types. It says a lot about high disagreement between real and declared intake.

 Unfortunately, because of the small sample size it is impossible to investigate if the differences between real and declared intake are related  to participants’ characteristics, especially – BMI.

 Nevertheless, the study is a further example how to deal with the development and implementation of dietary assessment tools and is valuable for the educational purposes.

Author Response

Dear reviewer,

Thank you for your feedback and useful comments. You will find below our answers to your comments.

Point 1: Is the research design appropriate? Can be improved

 Response 1: Our choice of methodology has been explained in the beginning of paragraph 2.2. before 2.2.1. (lines 74-76). We chose this methodology because the development of the food database was needed first considering the already existing structure of myfood24 which is an open-ended and online dietary assessment method.

Reviewer 2 Report

Dear authors, 

this manuscript  presents interesting and practical approach to improvement of nutritional assessment tools, much needed in field of public health nutrition. However, there are numerous parts of the text that need improvement regarding the language style, conceptualisation of the sentences, and grammar. Improvement in overall presentation of methodologies that were employed in the study are needed as well. Please, address all the highlighted and commented parts of the text.

Author Response

Dear reviewer,

Thank you for your feedback and useful comments. You will find below our answers to your comments.

Point 1: this manuscript  presents interesting and practical approach to improvement of nutritional assessment tools, much needed in field of public health nutrition. However, there are numerous parts of the text that need improvement regarding the language style, conceptualisation of the sentences, and grammar. […] Please, address all the highlighted and commented parts of the text.

Response 1: All comments have been addressed except for two:

  • It has been suggested that “J.-P. Blanc’s book’s” might be corrected into “J.-P. Blanc’s books” however only one book has been used (see reference no. 6)
  • The proposed reference in paragraph 4.2.1. has not been used as it does not assess the difference of perception according a food picture’s angle. The first sentence of the paragraph has been rephrased (lines 398-399) to make it clearer that the paragraph is about the difference of perception according to a picture’s angle.

Point 2: Improvement in overall presentation of methodologies that were employed in the study are needed as well

Response 2: Our choice of methodology has been explained in the beginning of paragraph 2.2. before 2.2.1. (lines 74-76). We chose this methodology because the development of the food database was needed first considering the already existing structure of myfood24 which is an open-ended and online dietary assessment method.